# Selective Inhibition of PDE4B Reduces Binge Drinking in Two C57BL/6 Substrains

**DOI:** 10.3390/ijms22115443

**Published:** 2021-05-21

**Authors:** C. Leonardo Jimenez Chavez, Camron D. Bryant, Melissa A. Munn-Chernoff, Karen K. Szumlinski

**Affiliations:** 1Department of Psychological and Brain Sciences, University of California Santa Barbara, Santa Barbara, CA 93106-9660, USA; cljc@ucsb.edu; 2Laboratory of Addiction Genetics, Department of Pharmacology and Experimental Therapeutics and Psychiatry, Boston University School of Medicine, Boston, MA 02118, USA; camron@bu.edu; 3Department of Psychiatry, University of North Carolina at Chapel Hill, Chapel Hill, NC 27599, USA; melissa_chernoff@med.unc.edu

**Keywords:** PDE4, alcoholism, drinking-in-the-dark, sedation, alcohol use disorder, intoxication, tolerance, binge drinking, C57 substrains

## Abstract

Cyclic AMP (cAMP)-dependent signaling is highly implicated in the pathophysiology of alcohol use disorder (AUD), with evidence supporting the efficacy of inhibiting the cAMP hydrolyzing enzyme phosphodiesterase 4 (PDE4) as a therapeutic strategy for drinking reduction. Off-target emetic effects associated with non-selective PDE4 inhibitors has prompted the development of selective PDE4 isozyme inhibitors for treating neuropsychiatric conditions. Herein, we examined the effect of a selective PDE4B inhibitor A33 (0–1.0 mg/kg) on alcohol drinking in both female and male mice from two genetically distinct C57BL/6 substrains. Under two different binge-drinking procedures, A33 pretreatment reduced alcohol intake in male and female mice of both substrains. In both drinking studies, there was no evidence for carry-over effects the next day; however, we did observe some sign of tolerance to A33’s effect on alcohol intake upon repeated, intermittent, treatment (5 injections of 1.0 mg/kg, every other day). Pretreatment with 1.0 mg/kg of A33 augmented sucrose intake by C57BL/6NJ, but not C57BL/6J, mice. In mice with a prior history of A33 pretreatment during alcohol-drinking, A33 (1.0 mg/kg) did not alter spontaneous locomotor activity or basal motor coordination, nor did it alter alcohol’s effects on motor activity, coordination or sedation. In a distinct cohort of alcohol-naïve mice, acute pretreatment with 1.0 mg/kg of A33 did not alter motor performance on a rotarod and reduced sensitivity to the acute intoxicating effects of alcohol. These data provide the first evidence that selective PDE4B inhibition is an effective strategy for reducing excessive alcohol intake in murine models of binge drinking, with minimal off-target effects. Despite reducing sensitivity to acute alcohol intoxication, PDE4B inhibition reduces binge alcohol drinking, without influencing behavioral sensitivity to alcohol in alcohol-experienced mice. Furthermore, A33 is equally effective in males and females and exerts a quantitatively similar reduction in alcohol intake in mice with a genetic predisposition for high versus moderate alcohol preference. Such findings further support the safety and potential clinical utility of targeting PDE4 for treating AUD.

## 1. Introduction

Cyclic AMP (cAMP)-dependent signaling has long been implicated in the neurobiology of alcohol and the neuropathology of alcohol use disorder (AUD) [1,2,3,4,5]. Cytosolic levels of cAMP are tightly regulated by certain families of cAMP-hydrolyzing phosphodiesterase (PDE) enzymes. In all, more than 100 isoforms of PDEs have been identified that are encoded by 21 different genes and categorized into families based on their structure and enzymatic activity. Of the 11 families, PDE4, PDE7 and PDE8 selectively deactivate cAMP [6,7], with PDE4 receiving considerable experimental attention in animal models of various neurological conditions, including affective disorders [8,9,10], neuroinflammation following injury [11], cognitive impairment/dementia [12,13,14,15], neurodegenerative diseases such as Alzheimer’s, Parkinson’s and Huntington’s diseases [16,17,18], schizophrenia [19] and substance use disorders, including psychostimulant use disorder [20,21,22,23,24] and AUD c.f., [3].

With respect to animal models of AUD, pretreatment with several non-selective PDE4 inhibitors (e.g., rolipram, Ro-20-1724, CDP840, mesopram, riclamilast) reduces alcohol preference drinking in traditional, continuous-access, 2-bottle-choice [25,26,27], as well as limited-access, binge-drinking paradigms [25,28]. Moreover, another non-selective PDE4 inhibitor, ibudilast, is reported to reduce alcohol intake in rat strains selectively bred for high alcohol intake/preference (i.e., P and HAD rats), and in high-alcohol-consuming/preferring C57BL/6J (B6J) mice [29]. Of all the non-selective PDE4 inhibitors tested in animal models of substance use disorder to date, ibudilast is the only one to advance to Phase II clinical trials for methamphetamine use disorder (ClinicalTrials.gov identifier: NCT01860807) and for oxycodone misuse (NCT01740414) and is currently in Phase 1 clinical trials for AUD (NCT02025998) [30], with recent reports indicating efficacy and safety [31,32,33,34,35,36,37].

The PDE4 family consists of 4 isozymes (PDE4A-D), all encoded by different genes [6,7]. These isozymes are difficult to distinguish pharmacologically because inhibitors target the active site on PDE4, and the absolute amino acid sequence of the active site is highly conserved across family members. Further related to the lack of specificity, most centrally acting PDE4 inhibitors (to include rolipram and ibudilast) have accompanying tolerability issues such as emesis and diarrhea that have limited their advancement in many clinical trials [38,39]. Accordingly, there has been a concerted effort to generate isozyme-selective PDE4 inhibitors as tools for studying their potential therapeutic and side effect profiles. Of particular interest to treating neuropsychiatric or neurophysiological conditions, PDE4A, PDE4B and PDE4D are expressed in the brain, notably within brain regions comprising the mesocorticolimbic system, which gates cognitive, affective and motivational processing [39,40,41]. Interestingly, PDE4A/B/D isozymes exhibit both distinct patterns of distribution in the brain and differences in subcellular localization. Gene knock-out (KO) studies in mice support their differential roles in behavior, including anxiety versus memory [42,43,44].

The advent of the PDE4B-specific inhibitor, A33 [45,46], enabled the study of selectively targeting the PDE4B isozyme in a reversible manner. A33 binds to a single amino acid in the C-terminus of PDE4B to promote the closing of this domain over the active site of the enzyme, thereby preventing access to cAMP and its hydrolysis [47,48]. This mechanism of inhibitory action confers a 50-fold greater selectivity of A33 for PDE4B (IC_50_ = 27 nM) over PDE4D (IC_50_ = 1569 nM) and other PDEs (IC_50_ > 10 μM), with a half-life of approximately 4 h in mouse brain [47,48]. As reported in Zhang et al. (2017) [46], acute administration of A33 to mice dose-dependently exerts anti-depressant-like effects in the forced swim and novelty-suppression of feeding tests, without altering anxiety-like behavior in the elevated plus maze or the marble-burying test, locomotor activity in an open field, novel object recognition or ketamine/xylazine-induced anesthesia (employed as a surrogate test for emesis in species that do not exhibit an emetic reflex [49]). In a rat model of traumatic brain injury, subchronic treatment with 0.3 mg/kg A33 ameliorated injury-induced heightening of contextual fear conditioning and deficits in spatial memory, without affecting injury-induced sensorimotor or working memory deficits [11]. While a recent study indicated that the novel PDE4B-selective inhibitor KVA-88-D reduced multiple measures of cocaine reinforcement in rats [50], to the best of our knowledge, no study has examined the effects of selective PDE4B inhibition in animal models related to AUD.

The present study employed adult (8–10 weeks of age) female and male C57BL/6NJ (B6NJ) mice to characterize the dose–response function for A33 effects upon binge alcohol drinking and to examine for the development of tolerance to the “anti-binge” effects of this inhibitor with repeated treatment. To increase the scientific rigor of the study, we conducted parallel alcohol-drinking studies in mice of the related, but genetically distinct, B6J substrain. The B6NJ and B6J substrains diverged nearly 70 years ago and exhibit different behavioral response to drugs of abuse e.g., [51,52,53,54,55], divergent binge eating phenotypes [56] and neuroimmune responses [57]. Relative to other unrelated mouse strains, both B6J and B6NJ mice readily consume alcohol under both continuous- [58] and limited-access drinking procedures [59]. Indeed, alcohol intake and preference is higher in B6J versus B6NJ mice [58,59,60,61] and only the B6J substrain escalates alcohol intake in response to a neuroimmune challenge [57]. Alcohol is well characterized to impact neuroimmune function [62,63,64]. As reviewed in detail elsewhere [65,66,67], by virtue of their ability to elevate cAMP levels and promote the activation of the cAMP signaolosome, PDE4 inhibitors can regulate both NFΚB- and Bcl-6-mediated induction of pro-inflammatory cytokines, the transcription of anti-inflammatory cytokines, T cell activation, the proliferation of other immune cells (e.g., macrophages) and neutrophil degranulation, and they augment the production of anti-inflammatory cytokines to impact the neuroimmune system. Thus, differences in drinking behavior between B6 substrains may influence the efficacy of neuroimmune modulators, such as selective PDE4B and non-selective PDE4 inhibitors, in reducing alcohol consumption [62,63,64]. As many non-selective PDE4 inhibitors exert sedative effects in rodents [68,69,70,71] and some are reported to potentiate the sedative–hypnotic properties of alcohol [72,73] (but see [26]), we also determined how the maximally effective A33 dose influenced motor function in both alcohol/A33-naïve and -experienced mice. We also examined the effect of this A33 dose on sucrose consumption to index general malaise.

Our results indicate that A33 effectively reduced binge alcohol intake in mice of both sexes and both substrains and did so under two different binge-drinking models, with some sign of tolerance developing to the effect of 1.0 mg/kg A33, when assayed under single-bottle binge drinking procedures. The “anti-binge” efficacy of 1.0 mg/kg A33 did not relate to a change in the sedative–hypnotic effects of alcohol in either alcohol-naïve or -experienced subjects, nor did it extend to sucrose consumption. Taken together, the results of this study support the potential clinical utility and relative safety of selectively targeting PDE4B to reduce excessive alcohol drinking.

## 2. Results

### 2.1. Experiment 1: Dose–Response Analysis of A33 Effects on Alcohol Drinking under Multi-Bottle-Choice Procedures, Followed by Assessment of A33 Effects on Sucrose Consumption, Spontaneous and Alcohol-Induced Changes in Motor Function in Alcohol/A33-Experienced Mice

To facilitate the navigation of Experiment 1 results, the procedural timeline for Experiment 1 is provided in Figure 1 below.

#### 2.1.1. A33 Reduces Alcohol-Drinking under Multi-Bottle-Choice Procedures in Both C57BL/6NJ and B6J Mice

Prior to the start of A33 pretreatment, B6NJ mice consumed on average 3.15 ± 0.51 g/kg of alcohol, whereas B6J mice consumed on average 5.17 ± 0.35 g/kg under our 3-bottle-choice procedures, a finding that aligns with prior comparisons of alcohol intake between these substrains [57,58,59,60,61]. There were no sex differences in the average baseline alcohol intake for either substrain (sex effects and sex X concentration interactions: for B6NJ, *p* > 0.016; for B6J, *p* > 0.20). Thus, the data for baseline alcohol intake were collapsed across sex to compare the dose–response function for alcohol intake between substrains. As illustrated in Figure 2A, B6NJ mice consumed equivalent amounts of alcohol from the 20% and 40% solutions prior to A33 administration (concentration effect: F(2,26) = 5.6, *p* = 0.009; 10% vs. 20%: t(14) = 2.73, *p* = 0.016; 20% vs. 40%: t(14) = 1.11, *p* = 0.29). In B6J mice, baseline alcohol intake increased as a function of alcohol concentration (Figure 2B), with mice consuming the majority of their total daily intake from the 40% alcohol solution (concentration effect: F(2,42) = 50.39, *p* < 0.0001; 10% vs. 20%: t(22) = 7.48, *p* < 0.0001; 20% vs. 40%: t(22) = 6.09, *p* < 0.0001). Thus, prior to A33 testing, B6J mice consumed more alcohol than B6NJ mice under our limited-access procedures, which reflected primarily higher intake of 40% alcohol.

Sex also did not affect the effect of A33 pretreatment on alcohol intake by either mouse substrain (for B6NJ, sex effect and interactions: *p* > 0.34; for B6J, sex effect and interactions: *p* > 0.08). Thus, the data were collapsed across sex for dose–response analyses of A33 effects on intake. Pretreatment with all A33 doses reduced the total alcohol intake by B6NJ mice (Figure 3A) (pretreatment effect: F(4,52) = 6.79, *p* < 0.0001; tests for simple main effects, VEH vs. 0.03 mg/kg: *p* = 0.04; vs. 0.1 mg/kg: *p* = 0.002; vs. 0.3 mg/kg: *p* = 0.004; vs. 1.0 mg/kg: *p* < 0.0001). Given the variability in baseline drinking exhibited by B6NJ mice under our multi-bottle DID procedure (Figure 2A), we determined whether or not the efficacy of A33 to reduce binge drinking might be predicted by individual differences in baseline intake. As expected, correlational analyses indicated a strong relationship between baseline intake and the intake of vehicle-pretreated B6NJ mice (r = 0.86, *p* < 0.0001; N = 15). However, there was no predictive relationship between baseline drinking and intakes observed following pretreatment with 0.03, 0.1 or 0.3 mg/kg of A33 (N = 15, 0.05 < r < 0.46, *p* > 0.09), although a positive correlation was detected for the 1.0 mg/kg dose of A33 (N = 15, r = 0.66, *p* = 0.008). This finding indicates that 1.0 mg/kg of A33 may be less effective in B6NJ mice with a higher baseline drinking.

An analysis of the alcohol dose–intake function in B6NJ mice indicated that the reduction in alcohol intake by A33 pretreatment varied by EtOH concentration (pretreatment X concentration: F(8104) = 3.48, *p* = 0.001). Deconstruction of the significant interaction along the concentration factor revealed no A33 effect on the intake of 10% alcohol by B6NJ mice (Figure 3B; dose effect: *p* = 0.87). A trend toward a significant reduction in the intake of 20% alcohol was observed in B6NJ subjects (Figure 3C; dose effect: F(4,56) = 2.32, *p* = 0.068), which reflected lower intake at the 0.1 and 1.0 mg/kg of A33 doses (tests for simple main effects, for 0.03 mg/kg: *p* = 0.06; for 0.1 mg/kg: *p* = 0.03; for 0.3 mg/kg: *p* = 0.122; for 1.0 mg/kg: *p* = 0.006). In contrast, A33 significantly reduced the intake of 40% alcohol in B6NJ mice (Figure 3D) (pretreatment effect: F(4,56) = 6.72, *p* < 0.0001) and this effect was observed at doses of 0.1 mg/kg and higher (tests for simple main effects, for 0.03 mg/kg: *p* = 0.09; for 0.1 mg/kg: *p* = 0.001; for 0.3 mg/kg: *p* = 0.005; for 1.0 mg/kg: 46, *p* = 0.001). These data provide novel evidence that inhibition of PDE4B is sufficient enough to reduce alcohol consumption in both female and male B6NJ mice.

Likewise, all A33 doses were effective at reducing the total alcohol intake exhibited by B6J mice (Figure 3E) (pretreatment effect: F(4,88) = 4.19, *p* = 0.004; tests for simple main effects, VEH vs. 0.03 mg/kg: *p* = 0.009; vs. 0.1 mg/kg: *p* = 0.15; vs. 0.3 mg/kg: *p* = 0.001; vs. 1.0 mg/kg: *p* = 0.003). However, despite the larger sample size, individual differences in baseline alcohol intake by B6J mice did not predict drinking when mice were pretreated with vehicle (r = 0.10, *p* = 0.67; N = 23) nor did it predict the efficacy of any A33 dose to reduce alcohol-drinking in this strain (Pearson’s correlational analyses, N = 23, −0.12 < r < 0.15, all *p* > 0.5).

Although the results of the statistical analysis indicated that the A33 effect in B6J mice did not vary with alcohol concentration (dose X concentration: *p* = 0.13), visual inspection of Figure 3F–H suggested otherwise. Thus, we examined the effects of A33 on the intake of each alcohol concentration separately, similar to what we investigated for B6NJ mice. No A33 dose altered the intake of 10% alcohol by B6J mice (Figure 3F; pretreatment effect: *p* = 0.91), while all doses significantly reduced their intake of 20% alcohol (Figure 3G) (pretreatment effect: F(4,88) = 6.63, *p* < 0.0001; tests for simple effects, VEH vs. 0.03 mg/kg: *p* = 0.002; vs. 0.1 mg/kg: 0.02; vs. 0.3 mg/kg: *p* < 0.0001; vs. 1.0 mg/kg: *p* = 0.001). However, only a statistical trend for an effect of A33 was detected on the intake of 40% alcohol by B6J mice (Figure 3H) (dose effect: *p* = 0.07; VEH vs. 0.03 mg/kg: *p* = 0.09; vs. 0.1 mg/kg: *p* = 0.05; vs. 0.3 mg/kg: *p* = 0.02; vs. 1.0 mg/kg: 0.04). These data extend the efficacy of A33 to reduce alcohol drinking to a mouse substrain with a high propensity to prefer and consume alcohol.

To test for carry-over effects, the average baseline alcohol intake was compared to that observed on each day following A33 administration. A main effect of day was observed for both B6NJ (F(5,70) = 2.97, *p* = 0.02) and B6J mice (F(5110) = 3.91, *p* = 0.005) (Figure 4). However, Bonferroni-corrected comparisons between baseline drinking and those observed following each of the five A33 injections (α = 0.0016) did not indicate any significant differences for either B6NJ (Figure 4A; *p* > 0.007, n.s.) or B6J mice (Figure 4B; *p* > 0.01, n.s.) or systematic change in drinking patterns (Figure 4). These data indicate that the effect of A33 upon alcohol intake is transient and not manifested the next day.

#### 2.1.2. A33 Differentially Affects Sucrose Intake in Alcohol-Experienced B6NJ vs. B6J Mice

In contrast to its effects on binge alcohol consumption, pretreatment with the maximum A33 dose (1.0 mg/kg) increased the intake of a highly palatable 20% sucrose by both male and female B6NJ mice (Figure 5A) (pretreatment effect: F(1,13) = 16.34, *p* = 0.001; sex effect and interaction, *p* > 0.78). In contrast, a sex-selective effect of 1.0 mg/kg of A33 was detected for sucrose intake in B6J mice (Figure 5B) (pretreatment X sex: F(1,21) = 6.21, *p* = 0.02). A33 pretreatment reduced sucrose intake by B6J females (t(11) = 2.46, *p* = 0.03), but not B6J males (Figure 5B; *t*-test, *p* = 0.32). While the data for B6J females indicate a potential for off-target effects, the fact that 1.0 mg/kg A33 did not reduce sucrose intake in B6J males and increased sucrose intake by B6NJ mice of both sexes shows that A33 does not universally induce malaise, nor does it blunt general reward processing or impair the ability to drink. Thus, the mechanisms underlying the reduction in alcohol intake in mice by the selective PDE4B inhibitor cannot simply be explained by an aversive state or by an overall dampening of reward processing.

#### 2.1.3. A33 Increases the Time Spent Stationary in A33/Alcohol-Experienced B6j Mice, But Does Not Alter Alcohol-Induced Locomotor Activity in Either Substrain

Female B6NJ mice exhibited greater spontaneous locomotor activity than males as indicated by a greater distance traveled and less time stationary during the 30 min immediately following VEH/A33 injection (Figure 6A) (sex effects, for distance traveled: F(1,13) = 9.43, *p* = 0.009; for time stationary: F(1,13) = 11.86, *p* = 0.004). However, A33 pretreatment (1.0 mg/kg) did not affect either spontaneous locomotor activity or stationary time in B6NJ mice of either sex during this period (Figure 6A; pretreatment effect and interactions, for distance traveled: *p* > 0.54; for time stationary: *p* > 0.40). In B6J mice, we detected no sex differences in spontaneous locomotion (sex effect, *p* = 0.74), although B6J females also spent less time stationary than males during the 30 min period immediately following VEH/A33 injection (Figure 6B) (sex effect: F(1,21) = 19.73, *p* < 0.0001). In contrast to B6NJ mice, A33 exerted sedative effects in B6J mice, as indicated by a non-significant trend towards lower locomotor activity (pretreatment effect: F(1,21) = 3.05, *p* = 0.10; sex X pretreatment: *p* = 0.43) and a significant increase in the time spent stationary, relative to VEH-pretreated controls (Figure 6B) (pretreatment effect: F(1,21) = 4.19, *p* = 0.05; sex X pretreatment: *p* = 0.21.

Following injection with 1.5 g/kg of alcohol, we detected no sex or pretreatment effects for either the distance traveled or time spent stationary by B6NJ mice (Appendix A; pretreatment X sex ANOVA: for distance traveled, all *p* > 0.31; for time stationary, all *p* > 0.11). Likewise, the alcohol-induced behavior of B6J mice was not significantly affected by A33 pretreatment (Appendix A; pretreatment X sex ANOVA: for distance traveled, all *p* > 0.26; for time stationary, all *p* > 0.13). Together, the locomotor activity results indicate that while A33 exerts sedative effects in B6J mice, the 1.0 mg/kg dose does not alter the psychomotor effects of a moderate dose of alcohol, at least in A33/alcohol-experienced mice.

#### 2.1.4. A33 Does Not Alter Baseline or Alcohol-Induced Reductions in Motor Coordination in A33/Alcohol-Experienced Mice

Compared to saline-injected controls, injection with 3 g/kg of alcohol markedly reduced the latency of female and male B6NJ (Figure 7A) and B6J mice (Figure 7B) to fall from the rotarod (for B6NJ; EtOH effect: F(1,12) = 314.26, *p* < 0.0001; EtOH X sex: *p* = 0.59; for B6J, EtOH effect: F(1,21) = 362.0, *p* < 0.0001; EtOH X sex: *p* = 0.26). Consistent with the results from the locomotor assay, pretreatment with 1.0 mg/kg of A33 did not alter basal motor co-ordination or alcohol-induced intoxication in either female or male B6NJ mice in Experiment 1 (Figure 7A; pretreatment effect and interactions, all *p* > 0.12). In this assay, no A33 effect was detected for the baseline or alcohol-induced reduction in the rotarod performance in the B6J mice of Experiment 1 (Figure 7B; pretreatment effect and interactions, all *p* > 0.44). Thus, at least in mice with a history of A33 and alcohol experience, A33 does not alter baseline motor coordination or alter the intoxicating properties of higher dose alcohol in either substrain.

#### 2.1.5. A33 Does Not Alter the Sedative–Hypnotic Effects of Alcohol in A33/Alcohol-Experienced Mice

Finally, we also failed to detect any effect of A33 pretreatment on the latency of either B6NJ (Figure 7C) or B6J mice (Figure 7D) following injection with 4 g/kg of alcohol (pretreatment X sex ANOVA, for B6NJ: all *p* > 0.19; for B6J: all *p* > 0.16). These final data further argue that A33 does not alter behavioral sensitivity to high-dose alcohol, at least in mice with a prior history of A33 and alcohol treatment.

### 2.2. Experiment 2: Examination of the Acute Effect of A33 on Motor Co-Ordination in Experimentally Naïve Mice, Followed by Assessment of the Effects of Repeated Treatment with A33 on Alcohol Consumption under Single-Bottle Procedures

To facilitate navigation of Experiment 2 results, the procedural timeline is provided in Figure 8 below.

#### 2.2.1. Acute Pretreatment with A33 Does Not Alter Rotarod Performance in Experimentally Naïve Mice

To address the possibility that the lack of any A33 upon spontaneous or alcohol-induced changes in motor behavior in Experiment 1 might reflect the development of tolerance to the effects of alcohol and/or A33, Experiment 2 commenced with a rotarod study conducted in experimentally naïve B6NJ and B6J mice. No sex difference was detected for either basal (sex effect and interactions, *p* > 0.12) or acute alcohol-induced changes in rotarod performance (sex effect and interactions, *p* > 0.09). Thus, the data for each variable were collapsed across sex and re-analyzed for the effects of A33. A pretreatment X strain ANOVA conducted for spontaneous rotarod performance indicated a significant interaction (F(1,23) = 5.17, *p* = 0.03). As illustrated in Figure 9A, this interaction reflected a non-significant reduction in the latency to fall in A33- versus VEH-pretreated B6J mice (*t*-test, *p* = 0.07), with no A33 effect observed in B6NJ mice (*t*-test, *p* = 0.30). As expected, based on our prior work [74,75], an acute injection with 2 g/kg of alcohol reduced the latency of mice to fall from the rotarod (Figure 9A,B), indicating that this dose induced intoxication.

Interestingly, acute pretreatment with 1.0 mg/kg A33 reduced the severity of this intoxication in both substrains, as indicated by a longer latency to fall, relative to VEH-pretreated mice (Figure 9B) (pretreatment effect: F(1,23) = 6.65, *p* = 0.02; strain effect and interaction, *p* > 0.10). Taken together, the results of this rotarod study argue that 1.0 mg/kg A33 does not induce motor incoordination and, if anything, reduces sensitivity to the intoxicating properties of alcohol.

#### 2.2.2. Repeated Pretreatment with A33 Does Not Produce Tolerance to Its Ability to Reduce Binge Drinking

One week following rotarod testing, the mice in Experiment 2 then underwent single-bottle DID procedures (20% *v*/*v* alcohol) to examine (1) whether or not the effect of A33 pretreatment on alcohol intake in Experiment 1 extends to the more traditional single-bottle DID model and (2) to address the possibility that tolerance might develop to the effect of A33 upon alcohol intake. Consistent with prior work [76], the total alcohol intake of both B6NJ and B6J mice prior to A33 pretreatment was lower when mice were offered only 20% alcohol (B6NJ: 2.10 ± 0.23 g/kg; B6J: 3.33 ± 0.75 g/kg), relative to that observed when mice were offered a choice between different alcohol concentrations (see Figure 2). Although it appeared that B6J mice also consumed, on average, more alcohol than B6NJ mice under single-bottle DID procedures, this substrain difference in intake was not statistically significant (strain effect and interaction, *p* > 0.12) and no sex difference in baseline alcohol intake was noted (sex effect, *p* = 0.26).

Next, we examined the acute effect of 1.0 mg/kg A33 on the intake of 20% alcohol by comparing the average total alcohol consumed prior to A33 pretreatment to that following the first injection. Replicating our results from Experiment 1, pretreatment with 1.0 mg/kg of A33 significantly reduced alcohol intake (pretreatment effect: F(1,20) = 17.86, *p* < 0.0001). Although the magnitude of the A33 effect appeared to be larger in B6J versus B6NJ mice (32% versus 22%, respectively; Figure 10A), this substrain difference was not supported by the results of the statistical analysis of these data (pretreatment X sex X strain ANOVA, all other *p* > 0.10). Having established that acute A33 pretreatment reduced intake in males and females of both substrains, we then examined whether the magnitude of the reduction varied as a function of repeated A33 treatment. Consistent with the results from Experiment 1 (Figure 4), the alcohol intake exhibited by both B6NJ and B6J was stable over the days between A33 injections (Figure 10C,D; substrain X sex X drinking day ANOVA, all *p* > 0.09). Thus, the alcohol intake following each A33 injection was expressed as a percent of the alcohol intake the day prior to more easily visualize any changes in the magnitude of the A33 effect. A comparison of the reduction in alcohol intake across the five A33 injections indicated a main A33 effect (F(4,80) = 4.77, *p* = 0.002), but no other main effects or interactions, (*p* > 0.44). As the magnitude of the A33 effect did not vary as a function of sex or substrain, the data were collapsed across these factors for follow-up analyses. As illustrated in Figure 10B, the significant pretreatment effect did not reflect a systematic change in A33 efficacy with repeated treatment, which was supported by the results of tests of within-subjects contrasts indicating a cubic pattern in the data (linear contrast: F(1,22) = 1.14, *p* = 0.30; cubic contrast: F(1,22) = 9.82, *p* = 0.005), with non-significant trends towards a larger A33 effect following the fourth injection, but a smaller effect following the fifth injection versus the first injection (Bonferroni-corrected *t*-tests, α = 0.003: injection 1 vs. 4, t(23) = 2.24, *p* = 0.04; 1 vs. 5: t(23) = 2.89, *p* = 0.008; 1 vs. 2 or 3, *p* > 0.62; all n.s.). Given the variability in alcohol intake under the single-bottle drinking procedure (Figure 10A), we determined whether or not individual differences in baseline alcohol intake predicted the efficacy of A33 pretreatment. Correlational analyses conducted between the average baseline alcohol intake and the efficacy of the first and last A33 pretreatment to reduce alcohol consumption indicated no predictive relationships (baseline vs. 1st A33 injection, r = −0.12, *p* = 0.58, N = 24; baseline vs. 5th A33 injection, r = −0.35, *p* = 0.1, N = 24). These latter results confirm that, regardless of baseline alcohol intake, pretreatment with 1.0 mg/kg A33 reduces alcohol intake under limited-access conditions, with no evidence for carry-over effects the next day. Further, these results suggest that some tolerance may develop to the acute effects of 1.0 mg/kg A33, at least when the drug is administered every other day.

Two days following the last A33 injection, trunk blood was collected to examine for group differences in BECs. On this day, we detected no group differences in alcohol intake (Figure 10E; strain X sex ANOVA, *p* > 0.19), and BECs correlated with intake (N = 24, r = 0.66, *p* < 0.0001) with an overall average BEC of 85.6 ± 12.3. Although the average BEC of B6NJ mice appeared to be higher than that of and B6J mice (Figure 10F; 95.2 ± 20.01 vs. 76.0 ± 14.5 mg/dL), this strain difference was not statistically significant (strain effect and interaction, *p* > 0.30). However, we did detect an overall sex difference in BECs (sex effect: (F1,23) = 7.65, *p* = 0.01) that reflected a higher BEC in male versus female mice (Figure 10F).

## 3. Discussion

Here, we show that pretreatment with the PDE4B isozyme-selective inhibitor A33 reduces binge alcohol drinking under two different procedures in B6NJ and B6J mice of both sexes. In A33/alcohol-experienced B6NJ mice, the A33-induced reduction in alcohol intake was unrelated to treatment-induced malaise, sedation or changes in behavioral sensitivity to alcohol. In comparably experienced B6J mice, A33 pretreatment reduced both sucrose intake by females and increased spontaneous immobility in both sexes. While such data indicate some off-target effects in the B6J substrain, A33 did not impair basal motor coordination in experimentally naïve or A33/alcohol-experienced mice of either substrain, nor did it alter behavioral sensitivity to a range of alcohol doses in A33/alcohol-experienced B6NJ or B6J mice. However, acute A33 pretreatment reduced behavioral signs of acute alcohol intoxication in mice of both sexes and substrains, suggesting that the capacity of A33 to lower alcohol intake may reflect a reduction, rather than an increase, in sensitivity to the intoxicating properties of alcohol. The present data align with a decade’s worth of evidence supporting the efficacy of non-selective PDE4 inhibitors for reducing alcohol consumption in various rodent models of AUD [25,26,27,28,29]. Our results also align with the outcomes of a Phase I clinical trial of ibudilast in humans with AUD, which demonstrated the tolerability of this non-selective PDE4 inhibitor, as well as its efficacy for reducing tonic craving, improving mood, and attenuating alcohol’s stimulant and mood-altering effects, in the absence of any ibudilast effects on the subjective response to alcohol [31]. While it remains to be determined whether ibudilast [31] or more isozyme-selective PDE4B inhibitors, such as A33 or KVA-88-D [50], will reduce alcohol consumption in humans, our data support the potential clinical utility of selective PDE4B inhibitors for reducing heavy drinking in AUD, with minimal off-target effects and no obvious interaction with alcohol, at least in alcohol-experienced subjects.

### 3.1. Substrain Differences in Binge Drinking under DID Procedures and Effects of A33 Upon Alcohol Consumption

Consistent with the limited literature comparing alcohol intake between B6 mice from the Jackson lineage versus the NIH lineage [57,58,59,60,61], alcohol intake under our 3-bottle, 2 h, DID procedure was greater in B6J versus B6NJ mice, with a similar (albeit less robust) substrain difference observed under a more traditional single-bottle DID procedure [77]. Although blood ethanol concentrations (BECs) could not be determined in Experiment 1 (see Methods), the levels of alcohol intake exhibited by B6NJ and B6J mice are predicted, based on our published data using this model [76,78,79,80] and those of others employing single-bottle DID procedures e.g., [77,81] to result in BECs below and above, respectively, the 80 mg/dL criterion for binge drinking [82]. Aligning with this position, although the alcohol intake was lower overall under the single- versus multi-bottle DID procedures employed in the present study, the overall average BEC attained under the single-bottle procedure was above the NIAAA criterion for binge drinking. Despite the substrain and procedure-related difference in alcohol intake, A33 pretreatment reduced alcohol consumption in both substrains under both drinking procedures. Thus, selective PDE4B inhibition is effective at reducing alcohol intake by non-preferring B6NJ mice with a moderate alcohol-drinking phenotype, as well as high-alcohol-preferring B6J mice with a heavy binge-drinking phenotype.

This being said, it is noteworthy that, in both experiments, alcohol intake returned to pre-A33 levels the day following pretreatment. Thus, the effects of A33 upon alcohol intake, at least under limited-access conditions, is transient, arguing against the notion that A33 pretreatment induces a taste-aversion to alcohol concentrations ranging from 10% to 40% (*v*/*v*). It is also noteworthy that a trend suggestive of the development of tolerance to the “anti-binge” effect of the 1.0 mg/kg dose of A33 was observed in both the B6NJ and B6J mice of Experiment 2. The total number of A33 injections administered in Experiment 2 was set at five to be consistent with the total number of injections received by the mice in the dose-response study of Experiment 1. Thus, it remains to be determined whether a more prolonged A33 treatment history, or a different temporal patterning of A33 delivery (i.e., daily or twice daily vs. every other day) would induce a more robust tolerance to the inhibitor’s “anti-binge” effects. It is also important to compare the relative efficacy of acute and repeated A33 treatment with that of other non-selective PDE4B inhibitors (e.g., ibudilast, ampremilast, rolipram). As our capacity to conduct research remains limited, such comparisons are planned for future work.

In Experiment 1, the effects of A33 on alcohol intake were dose-dependent in B6NJ mice, with the lowest and highest A33 doses reducing alcohol intake by 25% and 50%, respectively. Interestingly, the effect of A33 on total alcohol consumption in B6NJ mice primarily reflected a reduction in the intake of 40% alcohol. Thus, PDE4B inhibition blunts the rewarding properties of high-dose alcohol in B6NJ mice without producing a compensatory increase in the intake of lower alcohol concentrations. Notably, when only 20% alcohol was available for consumption (Experiment 2), the 1.0 mg/kg A33 dose was sufficient to reduce the intake of this concentration in B6NJ mice, indicating its efficacy in this B6 substrain under both multi- and single-bottle procedures. Although B6J mice consumed more alcohol than B6NJ mice in Experiment 1, the lowest dose of A33 (0.03 mg/kg) was similarly effective at reducing total alcohol intake in the B6J substrain (23% reduction). However, the A33 dose–intake function was relatively flat in the B6J mice, with the two highest doses reducing total alcohol intake by only 38%, with a similar magnitude of A33 effect observed with respect to the intake of 20% alcohol in Experiment 2. To avoid off-target effects, we limited the A33 dose-range of the present study to 1.0 mg/kg [46]. Whether higher A33 doses will be more effective at reducing binge alcohol intake by B6J mice and/or whether a greater A33 effect would be detected in B6J mice under continuous-access or an operant procedure remains to be determined but could inform the potential therapeutic utility of the A33 compound. Interestingly, the B6NJ substrain possesses 15 intronic single nucleotide polymorphisms, 1 intronic single base deletion, and 1 100 bp deletion in *Pde4b* relative to the B6J substrain [82,83,84], which might influence gene expression or splicing to impact alcohol intake and the capacity of A33 to curb excessive drinking.

### 3.2. Substrain by Sex Interaction in the Off-Target Effect of A33 on Sucrose Consumption

B6NJ mice are reported to binge-eat sucrose-laced food pellets under an intermittent, limited-access, place-conditioning procedure, while B6J mice do not [56]. However, when allowed 2 h of daily access to a 20% sucrose solution, we observed no obvious substrain difference in daily sucrose intake between B6NJ and B6J mice, although the pattern of drinking bouts was not assessed. Despite reducing the alcohol intake of both male and female B6NJ mice by half, pretreatment with 1.0 mg/kg of A33 increased sucrose consumption in these same mice. This finding was not entirely unexpected as A33 is reported to exert emetic-like effects in mice only at doses ≥10 mg/kg, with no emetic-like effects reported for 1.0 mg/kg of A33 [46]. Importantly, the fact that A33-pretreated B6NJ mice consumed more sucrose than controls argues against several potential mechanisms through which this inhibitor could reduce alcohol intake in this substrain, including malaise, a general disruption of reward processing, or the ability to drink or taste.

In contrast to B6NJ mice, administration of A33 at 1.0 mg/kg in B6J females produced a 20% reduction in sucrose consumption. This female-selective effect in B6J mice is peculiar given the aforementioned findings for B6NJ females, and may indicate a treatment interaction with B6J-specific gene variants (autosomal and/or sex chromosomes), or circulating ovarian hormones in determining sensitivity to what we presume to be A33-induced malaise in female B6J mice. In cardiac myocytes, cAMP levels are lower in females than males and this sex difference reflects a higher protein expression of PDE4B in females versus males [85]. In C57BL/6J mouse brain, no sex-based difference was reported in basal mRNA expression of different PDE4B slice variants (at least within the brain regions studied: hippocampal subregions, caudate putamen, cingulate cortex). However, the temporal patterning of PDE4B2 and PDE4B3 mRNA expression following a challenge of the innate immune system (e.g., lipopolysaccharide/LPS administration) is sex dependent, with females exhibiting a shorter latency to peak mRNA expression, a longer duration of high mRNA expression and a compensatory downregulation of mRNA expression that is not observed in males [86]. Recently, a substrain difference in the neuroimmune response to a challenge injection of polyinosinic:polycytidylic acid (poly(I:C)) was reported between B6NJ vs. B6J mice, with B6J mice exhibiting a smaller, but longer-lasting, innate immune response and only B6J mice exhibiting a poly(I:C)-induced proinflammatory response following alcohol consumption [57]. While purely circumstantial at this time, the fact that sex and substrain differences exist with respect to neuroimmune responsiveness involving PDE4B isozyme activity, means that it is possible that sex- and substrain-related factors may interact to influence behavioral sensitivity to PDE4B inhibitors and their off-target effects.

### 3.3. A33 Pretreatment Effects on Behavioral Sensitivity of Alcohol

PDE4B is also highly expressed in the dorsal striatum [40,41], raising the potential for off-target effects of inhibitors on basal ganglia function. In contrast to non-selective PDE4 inhibitors [26,70,71,72], neither constitutive *PDE4B* deletion [44] nor treatment with A33 in outbred ICR mice (0.1 to 3 mg/kg) [46] affected spontaneous locomotor activity. Consistent with this latter report, 1.0 mg/kg of A33 did not alter the distance traveled by the A33/alcohol-experienced B6NJ or B6J mice in Experiment 1. While one might argue that the failure to detect an A33 upon spontaneous motor behavior in Experiment 1 may reflect the development of tolerance to A33’s motor effects, acute pretreatment with 1.0 mg/kg of A33 did not alter the basal rotarod performance of the B6NJ and B6J mice of Experiment 2. However, in Experiment 1, the 1.0 mg/kg A33 dose increased the time spent stationary by both female and male B6NJ mice during the 30 min period following injection, without affecting the immobility in B6J mice. The B6NJ-selectivity of A33’s effect upon the time stationary was unexpected in light of our results for sucrose intake. Thus, it is not likely that the lower sucrose intake exhibited by B6J females reflects an A33-induced impairment in gross motor function, based on the results of our locomotor assay. To the best of our knowledge, this report is only the second to examine the effects of A33 on spontaneous mouse behavior. Thus, it remains to be determined whether our immobility result for B6J mice genuinely reflects a strain difference in behavioral sensitivity to A33, which may or may not be related to substrain differences in a *Pde4b* polymorphism or in innate neuroimmune function as discussed above. As suggested by (1) the largely negative results for motor behavior from Experiment 1, (2) the negative results for basal rotarod performance from Experiment 2 and (3) prior evidence from rat that the repeated administration (8–9 injections) of 0.3 mg/kg of A33 ameliorates motor impairments in a rat model of traumatic brain injury [11,87], the A33-induced increase in immobility expressed by B6NJ mice may simply be a spurious result, as the majority of findings argue little negative consequence of A33 treatment on gross motor function.

Systemic treatment with certain non-selective PDE4 inhibitors is reported to increase sensitivity to the sedative effects of alcohol [72,73], which reduces the margin of safety for this approach as a treatment for AUDs [3]. Despite increasing the time spent still by B6NJ mice when assayed in an alcohol-free state, A33 pretreatment (30 min prior) did not alter the subsequent locomotor response to 1.5 g/kg of alcohol in this substrain, nor did it affect alcohol-induced locomotion expressed by B6J mice in Experiment 1. Furthermore, A33 pretreatment did not alter alcohol-induced motor incoordination or sedation in either substrain in Experiment 1. The half-life of A33 ranges from 3.8 to 4.5 h in mouse brain [47,48]. Thus, the lack of an effect of A33 on these measures of alcohol-induced intoxication/sedation is unlikely to reflect insufficient dosing and/or the duration of testing. Supporting this notion, a 30 min IP pretreatment with 0.3 mg/kg of A33 resulted in brain levels of A33 in rat that are 4- to 7-fold higher than the IC_50_ against PDE4B3 measured in vitro [11,87]. Furthermore, herein, a 30 min pretreatment with A33 doses as low as 0.03 mg/kg was sufficient to reduce alcohol intake measured over a 2 h period (i.e., a testing period comparable to that of our locomotor activity and righting reflex assays).

Due to institutionally imposed limitations on animal numbers during the COVID-19 pandemic, Experiment 1 employed a within-subjects design in which each mouse received 7 injections of A33, spaced 3–5 days apart over the course of testing (Figure 1). This raises the possibility that tolerance developed to any A33-alcohol interactions that precluded detection of effects in the rotarod and righting reflex assays of Experiment 1. Indeed, in Experiment 2, A33 pretreatment increased the latency of both experimentally naïve B6NJ and B6J mice to fall from the rotarod upon the acute administration of 2 g/kg alcohol. This result is very interesting as it is opposite than that predicted from prior studies of non-selective PDE4 inhibitors [72,73] and indicates that despite producing no observable effect upon basal motor coordination, acute treatment with 1.0 mg/kg of A33 reduces, not potentiates, the intoxicating properties of alcohol. It will be important to determine in future work whether or not this acute A33 effect varies with repeated A33 treatment and/or alcohol history. However, the fact that A33 pretreatment reduced alcohol intoxication argues against an increase in the sedative–hypnotic properties of alcohol as the mechanism through which this inhibitor curbs alcohol intake. To the contrary, the present results suggest that A33 may exert its “anti-binge” effects by reducing sensitivity to alcohol’s subjective/intoxicating effects.

## 4. Materials and Methods

### 4.1. Subjects

Adult (8–10 weeks of age) female and male C57BL/6NJ mice (catalog no. 005304; Experiment 1: *n* = 15 with 8 females and 7 males; Experiment 2: *n* = 12 with 6 males and 6 females) and C57BL/6J mice (catalog no. 000664; Experiment 1: *n* = 23 with 12 females and 11 males; Experiment 2: *n* = 12 with 6 males and 6 females) were obtained from The Jackson Laboratory (Sacramento, CA, United States). Mice were housed in same-sex groups of 3–4 and allowed 7 days to acclimate to a climate- and humidity-controlled colony room, under a reverse 12 h light/dark cycle (lights off at 10:00 h). Mice were identified using tail markings. Food and water were available ad libitum except during the 2 h alcohol-drinking period (see below). All the cages were lined with sawdust bedding, with nesting materials and an igloo in accordance with vivarium protocols. All experimental procedures were in compliance with The Guide for the Care and Use of Laboratory Animals (2014) and approved by the Institutional Animal Care and Use Committee of the University of California, Santa Barbara (protocol number 829.3)

### 4.2. Drugs

A33 (CAS number 121604-72-6) was purchased from Tocris Bioscience (Minneapolis, MN, USA) and was suspended in 10% dimethyl sulfoxide (DMSO) at a concentration of 100 mg/mL and sonicated for 45 min. The suspension was then diluted using saline to the final concentrations of 0.03, 0.1, 0.3 and 1.0 mg/mL (0.1% DMSO), with the 1.0 mg/mL concentration requiring additional sonication for complete dissolution. The vehicle (VEH) solution consisted of 0.1% DMSO in saline. A33 pretreatment occurred 30 min prior to behavioral testing and was administered intraperitoneally (IP) at an injection volume of 10 mL/kg. As we [76] have demonstrated that mice tend to binge-drink more alcohol when offered a choice between various concentrations versus a single concentration alone, ethyl alcohol (190 proof) was diluted with potable tap water for consumption to final concentrations of 10%, 20% and 40% (*v*/*v*) for Experiment 1 as in recent binge-drinking studies by our group [78,79,80]. To determine whether or not the A33-induced reduction in alcohol intake under multi-bottle DID procedures generalizes to a more traditional single-bottle DID procedure, ethyl alcohol (190 proof) was diluted in potable water to a final concentration of 20% (*v*/*v*) [28,88,89,90,91]. For injection (1, 1.5, 2, 3 or 4 g/kg), ethyl alcohol (190 proof) was diluted in sterile saline and alcohol injections were administered at a larger, diluted volume of 20 mL/kg to reduce irritation at the injection site as in prior studies [74,75,92,93,94]. A procedural timeline of Experiment 1 is provided in Figure 1, while that for Experiment 2 is provided in Figure 8.

### 4.3. Experiment 1

#### 4.3.1. Multi-Bottle Drinking-in-the-Dark (DID) Procedures and A33 Pretreatment during Drinking 

Experiment 1 employed a 3-bottle DID procedure that we have shown repeatedly engenders consistently high alcohol intake (≥4.5 g/kg in 2 h) and high blood ethanol concentrations (BECs ≥ 80 mg/dL) in B6J isogenic and congenic mice [76,78,79,80]. Moreover, this multi-bottle-choice procedure enables an examination of A33 effects upon the dose–response function for alcohol binge drinking, important for conclusions regarding the potential psychopharmacological mechanisms through which A33 exerts its effects upon alcohol intake that are not afforded under the more traditional single-bottle DID procedure [28,77]. Group-housed mice were placed into individual, wire-top, drinking cages situated on a free-standing rack in the colony room at 2 h into the dark phase of the circadian cycle and allowed to habituate for at least 1 h. Mice were then presented with 3 sipper tubes containing unadulterated solutions of 10%, 20% and 40% (*v*/*v*) alcohol for a 2 h period, at which time the bottles were removed and the mice were returned to their home cages. The alcohol intake of each concentration during the session was determined by bottle weight before and after the 2 h drinking period and expressed as a function of body weight, which was determined on the days when mice were slated to receive an A33 injection. Spillage was monitored by placing bottles on an empty cage and the amount subtracted from each mouse’s intake on the corresponding day.

Following the establishment of a stable drinking baseline (4–7 days), Experiment 1 mice were randomly assigned to receive their first IP injection of one of five doses of A33 (0, 0.03, 0.1, 0.3 or 1.0 mg/kg). These doses are at, or below, those demonstrated to reduce floating behavior by ICR mice in a forced swim test, without affecting cognition, anxiety-like or emetic-like behavior [46]. Every 3–4 days, the mice in Experiment 1 were pretreated with a different dose of A33, using a Latin-square design. A within-subjects design was employed to increase the statistical power of the dose–response analyses given that the ongoing COVID-19 pandemic forced the Animal Resource Center of the University of California Santa Barbara to restrict the total animal numbers available for study. It is for this reason that the mice in Experiment 1 were also employed to assay for off-target effects as described in the subsections below.

#### 4.3.2. Sucrose Intake in A33/Alcohol-Experienced Mice

To examine for non-selective effects on fluid intake, a 20% sucrose solution was substituted for alcohol in Experiment 1 and the mice continued to drink for an additional 3 days (to allow for stabilization of intake) prior to pretreatment with either VEH or the maximum A33 dose from the alcohol-drinking phase of the study (1.0 mg/kg). The 20% sucrose concentration was selected as it engenders a high level of sucrose intake in B6J mice [95]. Three days later, the mice received the opposite pretreatment prior to sucrose presentation, in a cross-over, within-subjects design. As observed during the alcohol-drinking phase of the study, sucrose consumption returned to baseline levels the day following A33 injection. Sucrose intake was determined by the difference in bottle weight pre- and post-drinking, expressed as a function of body weight, which was determined prior to each A33 test day.

#### 4.3.3. A33 Effects Upon Basal and Alcohol-Induced Changes in Locomotor Activity, Intoxication and Sedation in A33/Alcohol-Experienced Mice

A33 was reported not to influence spontaneous locomotor activity in ICR mice [46]. However, its effects on alcohol-induced changes in motor behavior have not been determined. As a first pass examination of this issue, we next tested the mice from Experiment 1 for the effects of A33 pretreatment on locomotion induced by 1.5 g/kg alcohol. As in the sucrose study, we employed a cross-over experimental design in which mice were injected IP with either VEH or 1.0 mg/kg A33, with half of the mice from each sex receiving each pretreatment on the first test day. Mice were immediately placed into white polycarbonate activity chambers (25 cm W × 25 cm L × 30 cm H) and locomotor activity was monitored for 30 min by digital video-tracking (ANY-Maze; Stoelting Inc, Wood Dale, IL, USA). Following that 30 min period of baseline locomotor activity, mice were injected IP with 1.5 g/kg alcohol and behavior was recorded for an additional hour. The next day, the locomotor testing procedures were repeated but instead, mice received the alternate pretreatment.

The day following the locomotor testing, we employed a similar cross-over experimental design to examine the effects of pretreatment with 1.0 mg/kg of A33 on basal motor co-ordination and alcohol-induced motor incoordination in the rotarod assay. For this, Experiment 1 mice were first trained to walk on a fixed speed (10 rpm) rotarod (IIT Life Science, Woodland Hills, CA, USA), using procedures similar to those described previously [74,75,92]. Training commenced with an initial habituation to walking on the apparatus in a 3 min session, during which time the mice were placed immediately back onto the rotarod if they fell. Following habituation, mice underwent a series of 2 min training trials, during which time any mice that had fallen off of the rotarod remained on the floor of the apparatus until the next trial began. Once the mice could remain on the rotarod for three 2 min trials, they were considered to be “trained”. To determine the effects of A33 pretreatment on basal motor coordination, mice were then randomly assigned to receive an IP injection of either VEH or 1.0 mg/kg of A33. Thirty minutes later, mice underwent three consecutive 2 min test sessions and the latency to fall was recorded. Immediately following baseline testing under VEH/A33, mice were injected IP with 3.0 g/kg of alcohol and returned to the home cage for a period of approximately 45 min [92]. Then, mice underwent a second round of rotarod testing to examine the effect of A33 pretreatment alcohol intoxication. The next day, the rotarod procedures were repeated with animals receiving the alternate pretreatment.

In a final sub-experiment, we employed a cross-over design to examine the effects of A33 pretreatment on the latency of Experiment 1 mice to self-right following injection with 4 g/kg of alcohol [74,92]. Half of the mice were injected IP with VEH, while the other half received 1.0 mg/kg A33. Following pretreatment, mice were returned to their home cage. Thirty minutes later, mice were injected with 4 g/kg of alcohol and placed into an empty test cage. Within 3 min following the injection, mice were placed in a supine position and the time taken for the mice to turn over and place all four paws on the floor of the cage was determined. The next day, the experiment was repeated with mice receiving the alternate pretreatment.

#### 4.3.4. Statistical Analyses for Experiment 1

As the studies of B6NJ and B6J mice were conducted independently, spaced several months apart, the data were analyzed separately for each substrain. For both studies, the data for alcohol drinking were analyzed using mixed factor ANOVAs with sex as a between-subjects factor and pretreatment (5 levels) and EtOH concentration (3 levels) as within-subjects factors. Significant dose effects from the alcohol-drinking studies were subjected to post-hoc *t*-tests (when 1–3 comparisons were involved) or tests for simple main effects as appropriate for multiple comparisons. For the other variables, the data were analyzed using a sex by pretreatment ANOVA, with pretreatment as a within-subjects factor (2 levels). For the rotarod study, alcohol injection was also treated as a within-subjects factor (2 levels). Alpha was set at 0.05 for all initial analyses and adjusted for multiple comparisons when conducting post-hoc analyses.

### 4.4. Experiment 2

#### 4.4.1. A33 Effects Upon Basal Motor Co-Ordination and Alcohol Intoxication in Experimentally Naive Mice

As the mice in Experiment 1 were highly alcohol-experienced at the time of motor testing and had received multiple A33 injections, the lack of any A33 effect on the sedative–hypnotic properties of alcohol might reflect the development of tolerance to A33’s and/or alcohol’s motor effects. To address these possibilities, a rotarod experiment was first conducted in the Experiment 2 mice, one week prior to commencing drinking procedures (i.e., when the mice were experimentally naïve). Experiment 2 mice were habituated to, and trained to walk on, the rotarod as described in Section 4.3.3 above. Then, half of the mice from each sex and substrain were pretreated with VEH or 1.0 mg/kg A33 (*n* = 6/substrain/sex; again, animal numbers restricted due to on-going pandemic). Thirty minutes later, mice underwent three 2 min tests to examine for the effect of acute A33 pretreatment upon basal motor coordination. Immediately following this test, all mice were injected with 2 g/kg alcohol. Thirty minutes later, mice were tested again for rotarod performance.

#### 4.4.2. Single-Bottle DID Procedures and A33 Pretreatment during Drinking

A week following rotarod testing, we next assayed the effects of repeated pretreatment with 1.0 mg/kg A33 upon alcohol-induced under more traditional single-bottle DID procedures [27,75]. The procedures for habituating the mice to the drinking cages and sipper tube presentation were identical to those described in Section 4.3.1 above, with the exception that a single sipper tube containing 20% (*v*/*v*) alcohol was presented for the 2-h period and alcohol intake was based solely on the volume consumed from the single sipper tube. Alcohol intake under this procedure stabilized in both substrains within 4 days. Following intake stabilization, all mice were pretreated with 1.0 mg/kg A33 (the maximum dose tested in Experiment 1), 30 min prior to the drinking session to establish the acute effect of A33 upon alcohol intake under the single-bottle DID procedure. As detailed in the Results for Experiment 2, alcohol intake returned to pre-A33 levels the day following the first injection, providing further evidence for no overt carry-over effects upon alcohol intake. Thus, mice in Experiment 2 were pretreated with 1.0 mg/kg A33, every 2–3 days, to examine for the development of tolerance to the reduction in intake. In a manner consistent with the design of the drinking study in Experiment 1 (5 doses), mice were pretreated with a total of five injections of 1.0 mg/kg A33.

#### 4.4.3. Statistical Analyses for Experiment 2

As the studies of B6NJ and B6J mice were conducted independently, spaced several months apart, the data were analyzed separately for each substrain. For both studies, the data for alcohol drinking were analyzed using mixed factor ANOVAs with sex as a between-subjects factor and pretreatment (5 levels) and EtOH concentration (3 levels) as within-subjects factors. Significant dose effects from the alcohol-drinking studies were subjected to post-hoc *t*-tests (when 1–3 comparisons were involved) or tests for simple main effects as appropriate for multiple comparisons. For the other variables, the data were analyzed using a sex by pretreatment ANOVA, with pretreatment as a within-subjects factor (2 levels). For the rotarod study, alcohol injection was also treated as a within-subjects factor (2 levels). Alpha was set at 0.05 for all analyses and adjusted for multiple comparisons when conducting post-hoc analyses.

### 4.5. Blood Sampling and BEC Analysis

For both experiments, blood samples (50 μL) were collected 1–2 days following the last drinking session (i.e., following the last A33 injection). All the blood samples were collected in blood collection tubes lined with lithium heparin (BD Vacutainer, Mississauga, ON, Canada) and centrifuged at 10,500 rpm at 4 °C for 20 min to obtain plasma. The extracted plasma sample was kept frozen at −80 °C until assayed by headspace gas chromatography. BECs were determined using a Shimadzu GC-2014 gas chromatography system (Shimadzu, Columbia, MD, USA) and the data were determined via the GC Solutions version 2.10.00 software (Aligent, Santa Clara, CA, USA). Samples were diluted at 1:9 with non-bacteriostatic saline (50 μL of sample). Toluene was used as the pre-solvent due to its lower boiling point versus ethanol. Each sample was tested within 1 week of blood collection to reduce the potential for ethanol evaporation during storage. The determination of ethanol from each sample was derived using the standard curve equation determined prior to analyses of the samples. A new standard curve was formulated for each cohort of blood samples to ensure maximal accuracy. After the ethanol peak area was determined, the peak area was used to determine the ethanol concentration and subsequently the percent of ethanol in the blood. The BECs were then correlated with the alcohol intake observed on the day of blood sampling.

## 5. Conclusions

The PDE4B-selective inhibitor A33 dose-dependently reduces binge alcohol drinking in two B6 substrains that vary in their alcohol preference and intake. The capacity of A33 to curb binge drinking did not relate in any obvious manner to the effect of pharmacological inhibition on sucrose intake, baseline motor activity or coordination or to a change in sensitivity to alcohol’s sedative–hypnotic effects. Such findings further support the potential clinical utility of targeting PDE4 for treating AUD and indicate that selective inhibition of the PDE4B isozyme may be an effective and safer strategy than non-selective inhibitors for therapeutic intervention.

## Figures and Tables

**Figure 1 ijms-22-05443-f001:**
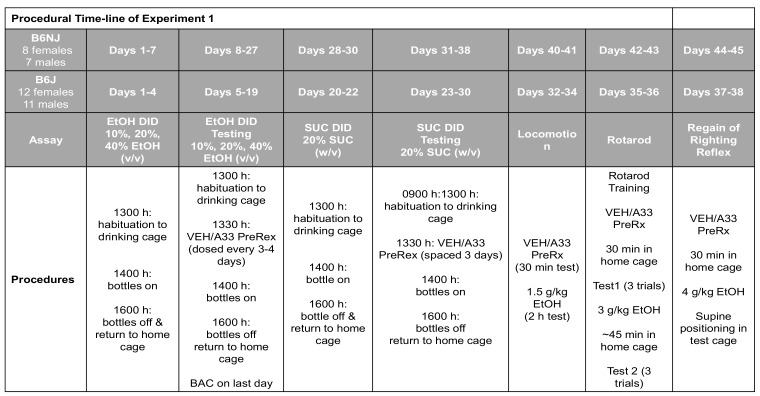
Summary of the procedural timeline for Experiment 1, which employed a within-subjects design to determine (**1**) the dose–response function for A33 effects upon alcohol (EtOH) intake under 3-bottle drinking-in-the-dark (DID) procedures; (**2**) the effect of 1.0 g/kg A33 on sucrose (SUC) intake under DID procedures; and (**3**) the effect of 1.0 mg/kg A33 tests for spontaneous and alcohol-induced changes in locomotor activity, rotarod performance and then alcohol-induced sedation using the regain of righting reflex assay.

**Figure 2 ijms-22-05443-f002:**
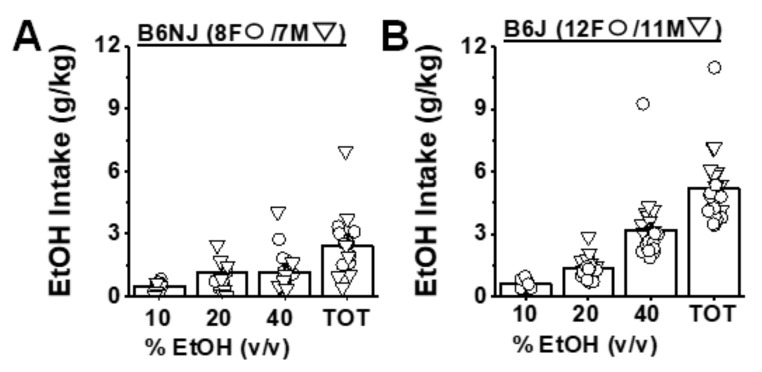
Baseline alcohol intake under 3-bottle-choice procedures in B6NJ and B6J mice. Summary of the dose–response function for alcohol (EtOH) intake and the total (TOT) alcohol intake of female (F) and male (M) (**A**) B6NJ and (**B**) B6J mice prior to A33 testing. The data represent the mean ± SEM of the number of mice indicated in each panel, in addition to the individual data points for each concentration and total intake.

**Figure 3 ijms-22-05443-f003:**
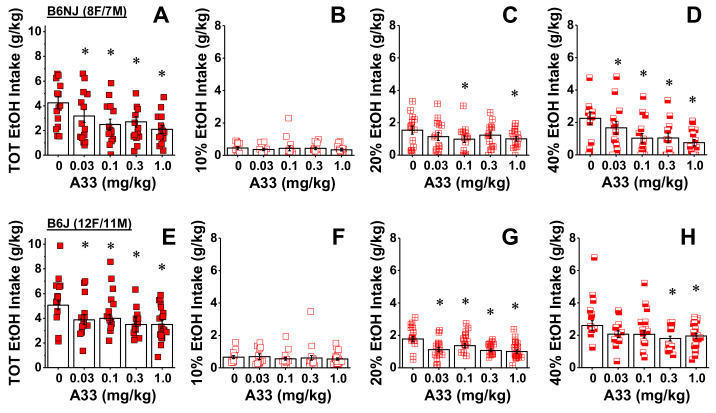
A33 reduces binge alcohol intake in both B6NJ and B6J mice. Summary of the dose–response function for A33-induced reduction in (**A**) total alcohol (EtOH) intake and (**B**–**D**) intake from the sipper tubes containing, respectively, 10%, 20% and 40% alcohol in female and male B6NJ mice (8 females/7 males). (**E**) Same as Panel A for female and male B6J mice. (**F**–**H**) Same as Panels B–D for female and male B6J mice (12 females/11 males). The data represent the mean ± SEM of the number of mice indicated, in addition to the individual data points for each A33 dose. * *p* < 0.05 vs. 0 mg/kg of A33 (tests for simple main effects).

**Figure 4 ijms-22-05443-f004:**
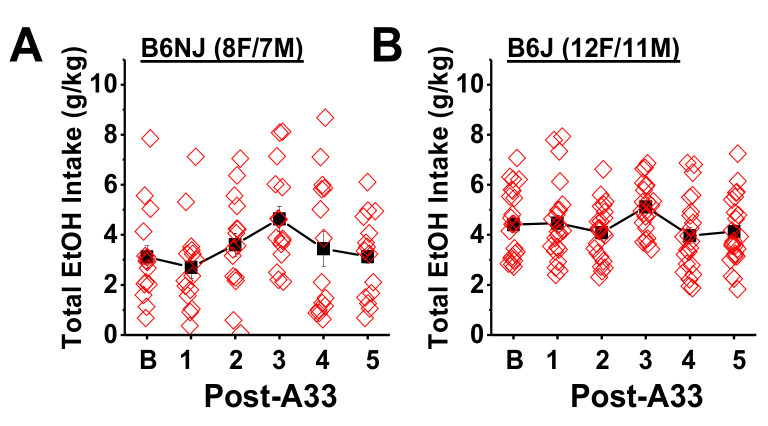
A33 dosing does not produce any carry-over effects the day following injection. Comparison of the average baseline alcohol intake exhibited by (**A**) B6NJ and (**B**) B6J mice and the total alcohol intake exhibited the day following each A33 injection (Post-A33). The data represent the mean ± SEM of the number of mice indicated in parentheses, as well as the individual data points for each drinking day.

**Figure 5 ijms-22-05443-f005:**
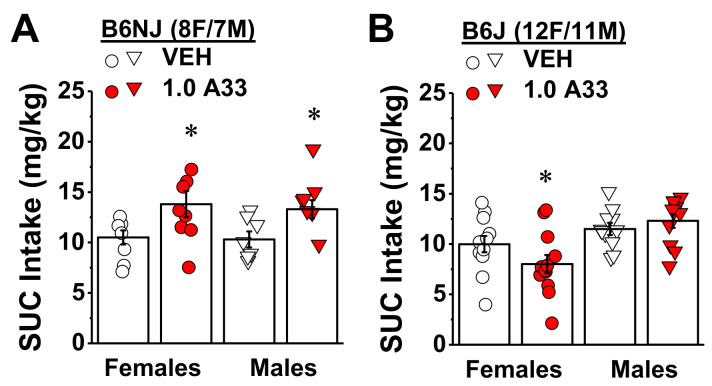
Strain by sex interaction in the effect of A33 on sucrose intake. Summary of the effect of pretreatment with either vehicle (VEH) or 1.0 mg/kg A33 (1.0 A33) on the intake of a 20% sucrose (SUC) solution under a 2 h DID procedure in (**A**) female and male B6NJ mice (8 females/7 males) and (**B**) female and male B6J mice (12 female/11 male). The data represent the mean ± SEM of the number of mice indicated, in addition to the individual data points for each experimental group. * *p* < 0.05 vs. 0 mg/kg of A33 (main effect).

**Figure 6 ijms-22-05443-f006:**
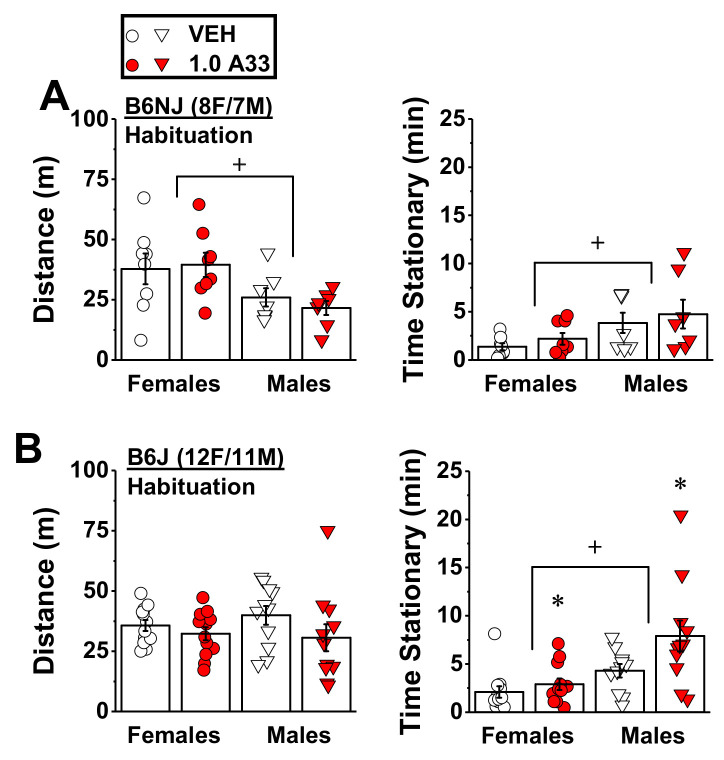
Effect of A33 on spontaneous locomotor activity. Summary of the effects of immediate pretreatment with either vehicle (VEH) or 1.0 mg/kg of A33 (1.0 A33) on (**A**) the total distance traveled and the time spent stationary by female and male B6NJ mice during a 30 min test (8 females/7 males). (**B**) Same as Panel A for female and male B6J mice (12 females/11 males). The data represent the mean ± SEM of the number of mice indicated, in addition to the individual data points for each experimental group. * *p* < 0.05 vs. 0 mg/kg of A33 (main effect); + *p* < 0.05 vs. females (main effect).

**Figure 7 ijms-22-05443-f007:**
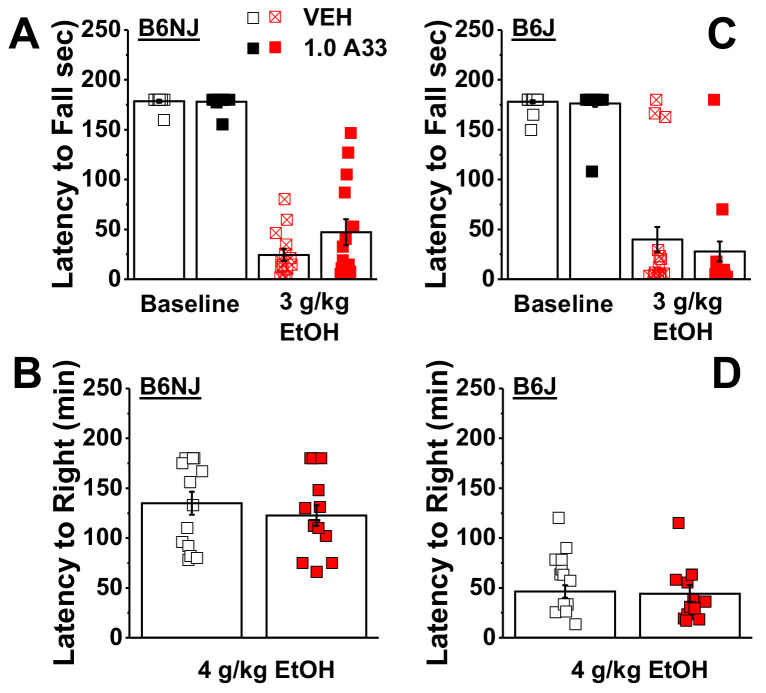
A33 does not affect motor coordination or the sedative–hypnotic properties of alcohol in A33/alcohol-experienced mice. Summary of the effects of pretreatment with either vehicle (VEH) or 1.0 mg/kg of A33 (1.0 A33) on (**A**) the latency of female and male B6NJ mice (8 females/7 males) to fall from a rotarod under an alcohol-free (Base; 30 min pretreatment) and alcohol-intoxicated state (3 mg/kg EtOH; ~75 min pretreatment). (**B**) Same as Panel A for female and male B6J mice (12 females/11 males). (**C**) Summary of the effects of VEH or A33 upon the time taken by B6NJ mice to regain their righting reflex following injection with 4 g/kg of alcohol. (**D**) Same as Panel C for B6J mice. The data represent the mean ± SEM of the aforementioned number of mice, in addition to the individual data points for each experimental group.

**Figure 8 ijms-22-05443-f008:**
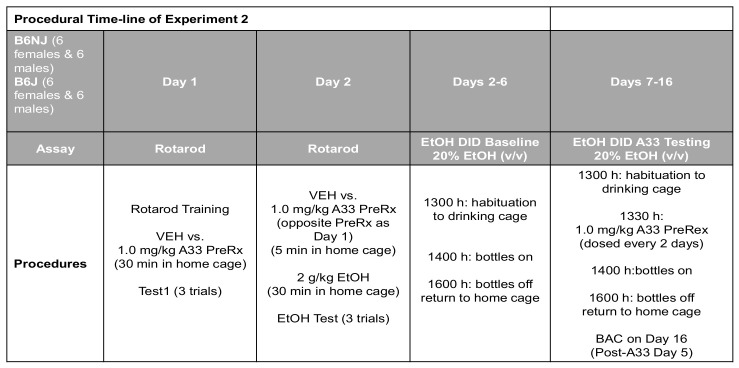
Summary of the procedural timeline for Experiment 2, which first determined the effect of 1.0 mg/kg A33 upon basal and alcohol-induced changes in rotarod performance in experimentally naïve mice, followed by an examination of the effect of repeated pretreatment with 1.0 mg/kg A33 upon alcohol (EtOH) intake under a single-bottle drinking-in-the-dark (DID) procedures.

**Figure 9 ijms-22-05443-f009:**
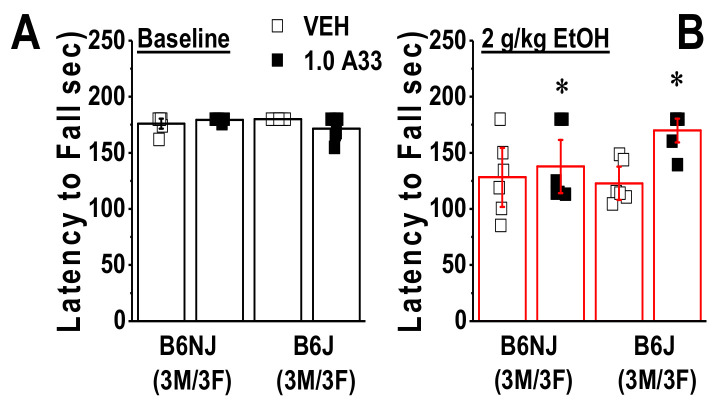
A33 does not affect basal motor coordination, but reduces the intoxicating properties of 2 g/kg alcohol in experimentally-naïve mice. Summary of the effects of pretreatment with either vehicle (VEH) or 1.0 mg/kg of A33 (1.0 A33; 30 min prior) on (**A**) the latency of B6NJ and B6J mice (3 females/3 males/group) to fall from a rotarod under an alcohol-free state and (**B**) following injection with 2 g/kg EtOH (~30 min post-alcohol). The data represent the mean ± SEM of the number of mice indicated, in addition to the individual data points for each experimental group. * *p* < 0.05 vs. VEH (main effect).

**Figure 10 ijms-22-05443-f010:**
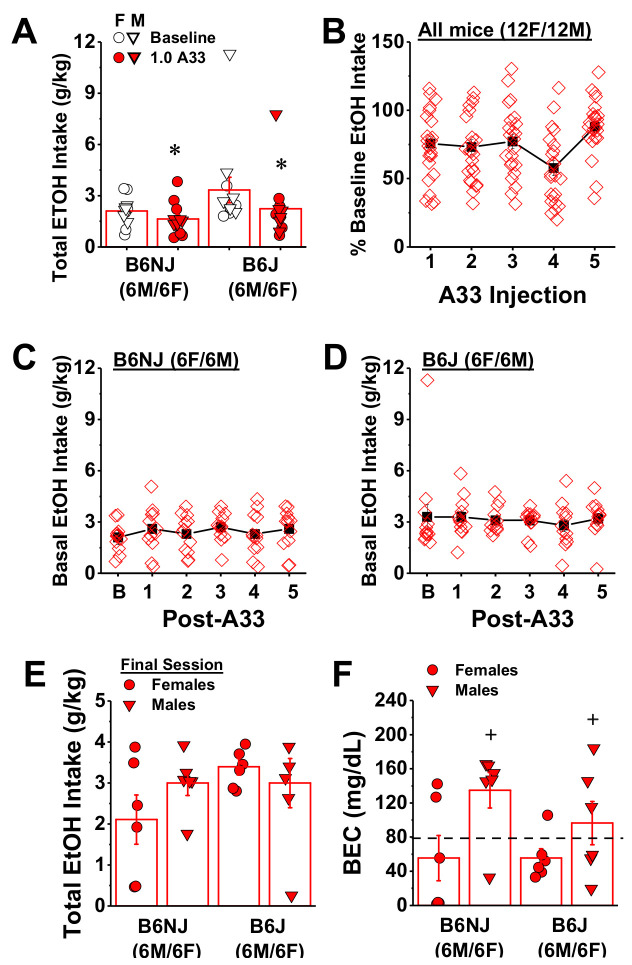
Repeated, intermittent treatment with 1.0 mg/kg A33 reduces alcohol intake under single-bottle DID procedures. (**A**) Comparison of the effects of acute pretreatment (30 min prior) with 1.0 mg/kg of A33 (1.0 A33) upon the intake of 20% alcohol under DID procedures between B6NJ and B6J mice. (**B**) Summary of the effect of repeated, intermittent, pretreatment with 1.0 mg/kg of A33 upon the intake of 20% alcohol by all of the mice in Experiment 2. Note that the data are expressed as a percentage of the average baseline alcohol intake, depicted in Panel A. Comparison of the baseline intake of 20% alcohol prior to A33 treatment (**B**) and the alcohol intake observed on the days between A33 injections in (**C**) B6NJ and (**D**) B6J mice. Comparison of the alcohol intake exhibited by female and male B6NJ and B6J mice on the final drinking day prior to blood collection (**E**) and their resultant BECs (**F**). The data represent the mean ± SEM of the number of mice indicated, in addition to the individual data points for each experimental group. * *p* < 0.05 vs. Baseline (main effect); + vs. Females (main effect).

## Data Availability

The data presented in this study are available on request from the corresponding author.

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
