# Peer review of "Selective Inhibition of PDE4B Reduces Binge Drinking in Two C57BL/6 Substrains"

_ijms, 2021, doi:10.3390/ijms22115443_

Round 1

Reviewer 1 Report

Chavez et al present a series of studies testing whether a selective inhibitor of phosphodiesterase 4 modulates binge drinking behaviors in mice. Although the experiments rely on a single small molecule inhibitor A33, the dose, pretreatment time, and route are justified well by pharmacokinetics studies cited in the Introduction. Here, the authors find that A33 decreases voluntary drinking in two models in two strains of mice. The results from a thorough series of subsequent studies suggests that A33 may decrease drinking by altering the intoxicating or rewarding properties of ethanol, without affecting sedative-hypnotic properties or general caloric rewards. The experiments are well-controlled and appear technically sound. The authors should be commended for designing efficient longitudinal studies during COVID-19 restrictions and for their transparency in detailing the study design and disclosing related limitations. Nonetheless, prior to publication, I have two points and several comments that I think would improve the manuscript.

Major points

  • There is considerable variability in drinking in both strains in the multi-bottle DID procedure. Was A33 efficacy related to the baseline drinking phenotype? E.g. did A33 reduce the amount of ethanol consumed similarly in high-drinking individuals versus low-drinking individuals?
  • It is somewhat surprising that sex differences in ethanol DID were not observed, considering literature that female mice voluntarily drinking more than male counterparts. Data from female and male mice should be presented separately in supplemental figures or denoted with distinct symbols in the main figures.

Minor comments and line edits

  • Some clarification regarding the statistical tests used is needed. How were post-hoc tests corrected for multiple comparisons? Line 143, is “p’s>0.016” a typographical error? If not, that would seem to imply that some of the comparisons reached significance (p<0.05). A similar instance occurs on line 210 and elsewhere. Also, for Figure 5, the markings in the legend do not match those in the figure (‘+’ in legend but only asterisks in the figure).
  • In Figure 1A, the bar does not visually appear to be located at the mean of the symbols in the total column, can the authors double-check this plot?
  • Why was the concentration of sucrose selected to be 20%?
  • In my opinion, “Time immobile” is an unusual way to describe a behavior in an open field locomotor assay, as it often signifies helplessness-like behavior in forced swim or tail suspension tests. I would suggest replacing this term with “stationary time” or “rest time”.
  • Figure 6 – It is a bit hard to follow the timing of the assay as these data are presented. Half of these data appear to be taken from a 30-minute habituation period whereas other panels are from a 2-hour segment. A more straightforward comparison, like A33 versus saline treatments over time, would be easier to follow.
  • The methods section mentions that blood was collected for the second experiment and BECs were assessed, but I do not see these data included in the manuscript.
  • Line 57. “I.e.” instead of “e.g.”?
  • Line 96, “related to AUD” instead of “of AUD”
  • Line 97, include “greater than” in age statement
  • Line 109, how is a PDE4 inhibitor a neuroimmune modulator?
  • Line 203, “These data … indicate that higher A33 doses are required to reduce high-dose alcohol consumption in the higher-drinking B6J substrain.” I would remove this claim as the dose-response curves overall appear comparable in these two strains. Also, in this study, there does not appear to be a baseline strain difference in drinking between the vehicle-treated B6NJ and B6J mice.
  • Line 404 – A33 is not a “medication”
  • Line 435 – should read “80 mg/dL”

Author Response

We would like to begin by sincerely thanking this reviewer who clearly spent a great deal of time reviewing our report.  They noticed several very glaring errors and omissions and for that, we are very grateful. 

1) There is considerable variability in drinking in both strains in the multi-bottle DID procedure. Was A33 efficacy related to the baseline drinking phenotype? E.g. did A33 reduce the amount of ethanol consumed similarly in high-drinking individuals versus low-drinking individuals?

Reply: We agree that alcohol intake was relatively variable in both studies – not just in the study employing the multi-bottle choice procedure. We appreciate the reviewer’s suggestion to try to relate individual differences in baseline drinking to the effectiveness of A33 and we conducted the suggested correlational analyses.  As now indicated in lines 175-184, baseline drinking by B6NJ mice was correlated with drinking under both 0 and 1.0 mg/kg A33, but not under the other A33 doses. No correlations were found for A33 and B6J mice (lines 212-216). In the second study, no correlations were seen between baseline drinking and the % reduction in intake by A33. Collectively, these data do not strongly support a predictive relationship between baseline drinking and A33’s effectiveness to reduce drinking.

2) It is somewhat surprising that sex differences in ethanol DID were not observed, considering literature that female mice voluntarily drinking more than male counterparts. Data from female and male mice should be presented separately in supplemental figures or denoted with distinct symbols in the main figures

Reply:  We were equally surprised by the lack of a sex difference in alcohol intake, particularly has we have observed sex differences under 1, 2, 3 and 4-bottle DID procedures in the past.  We have no explanation as to why the female B6J or B6NJ mice did not consume more alcohol than their male counterparts.  Female mice currently drinking in the laboratory are exhibiting higher binge-drinking than males.  As our ability to be in the building was highly restricted during these experiments, it is impossible to know whether or not extraneous factors (e.g., noise) might have negatively impacted the intake of the female mice. We have taken the reviewer’s suggestion and now indicate using different symbols who is male versus female in the figures for baseline drinking (Fig. 2 and Fig. 10A).  From these plots, one can readily see that the variability occurred in both male and female subjects, with no systematic separation in the amount of alcohol consumed. In the interest of reducing the complexity of the graphs of A33’s effects, we opted not to present the data separately for males and females where there was no indication of any sex effect of interaction.   

3) Some clarification regarding the statistical tests used is needed. How were post-hoc tests corrected for multiple comparisons? Line 143, is “p’s>0.016” a typographical error? If not, that would seem to imply that some of the comparisons reached significance (p<0.05). A similar instance occurs on line 210 and elsewhere. Also, for Figure 5, the markings in the legend do not match those in the figure (‘+’ in legend but only asterisks in the figure).

Reply: When a Bonferroni correction was applied, we changed the alpha setting from 0.05 according to the number of comparisons.  So, for 4 comparisons (i.e., 0 vs 0.03, 0 vs. 0.1, 0 vs. 0.3 and 0 vs. 1.0 A33, we divided 0.05 by 4 to yield a new alpha of 0.0125.  So, in order for a group difference to be significant, p<0.0125 and not p<0.05.  Thus, a p=0.016 is higher than p=0.0125 and the group difference is not statistically significant. This is the traditional method to apply a Bonferroni correction for multiple comparisons. We realized that we failed to mention that we applied corrections to our post-hoc analyses and this is now clearly stated in the Methods section.

We have corrected the figure legend for Figure 5 as there are no “+”.  Thank you for noticing this error.  

4) In Figure 1A, the bar does not visually appear to be located at the mean of the symbols in the total column, can the authors double-check this plot?

Reply:  Thank you for noticing this error, the total column mean definitely does not match the spread of the data!  We have corrected the mean in the TOT column.  Again, thank you for noticing this copy and paste error when graphing.

5) Why was the concentration of sucrose selected to be 20%?

Reply:  This concentration of sucrose is higher than that employed in our prior work focused on binge-drinking (where we employ 5 or 10% solutions); however, we wanted to engender a high level of intake and this concentration is preferred by B6J mice (see new Ref. 104). This rationale is now provided in the Methods.

6) In my opinion, “Time immobile” is an unusual way to describe a behavior in an open field locomotor assay, as it often signifies helplessness-like behavior in forced swim or tail suspension tests. I would suggest replacing this term with “stationary time” or “rest time”.

Reply:  We understand the reviewer’s concern.  The term “immobility” is derived from the digital tracking program and this is why it was employed originally.  As we do not know if the mice are “resting” (which, in my opinion, implies that they are taking a break from physical activity or are tired), we have opted to change “immobility time” to “stationary time”.

7) Figure 6 – It is a bit hard to follow the timing of the assay as these data are presented. Half of these data appear to be taken from a 30-minute habituation period whereas other panels are from a 2-hour segment. A more straightforward comparison, like A33 versus saline treatments over time, would be easier to follow.

Reply: The data were presented in columns in order to better visualize the means with the scatterplots, which would be even more confusing and cluttered if presented as a time-course.  We completely agree that the original Figure 6 is very busy.  As we detected no effects of A33 on locomotor activity (and we really didn’t see much in the way of alcohol-induced locomotion), we have opted to move these figures to the supplement and they are now separated into two distinct supplemental figures. 

8) The methods section mentions that blood was collected for the second experiment and BECs were assessed, but I do not see these data included in the manuscript.

Reply: Our sincere apologies and thank you for pointing this out.  Somehow that entire section ended up edited out of the submitted manuscript!  We have now included the information in lines 440 to 447 and the data are now presented in panels E and F of Figure 10.

Other minor issues:

Line 57. “I.e.” instead of “e.g.”?

Reply:  corrected

Line 96, “related to AUD” instead of “of AUD”

Reply: corrected

Line 97, include “greater than” in age statement

Reply:  We could not locate this “age statement” in or around line 97.

Line 109, how is a PDE4 inhibitor a neuroimmune modulator?

Reply: We have now included a brief description in the introduction regarding the many mechanisms involved in PDE4 regulation of neuroimmune function. See lines 113-119.

Line 203, “These data … indicate that higher A33 doses are required to reduce high-dose alcohol consumption in the higher-drinking B6J substrain.” I would remove this claim as the dose-response curves overall appear comparable in these two strains. Also, in this study, there does not appear to be a baseline strain difference in drinking between the vehicle-treated B6NJ and B6J mice.

Reply:  We have removed this sentence.

Line 404 – A33 is not a “medication”

Reply:  We have changed to “treatment”

Line 435 – should read “80 mg/dL”

Reply:  Thank you for catching that, we have corrected.

Reviewer 2 Report

The manuscript describes results of a preclinical mouse study on the effects of a selective phosphodiesterase PDE4B inhibitors on alcohol drinking in a binge drinking model. This study builds on earlier finding of other researchers on the effects of other PDE inhibitors and extends the findings to this selective subclass. The inhibition of alcohol drinking was transient, showing no carry over to the next session. Using two C57 strains it was shown that the drug at alcohol-drinking reducing dose did not alter consistently alcohol intoxication or sensitivity, but affected sucrose drinking (increased in one strain!)

While the manuscript is well written and understandable, with proper statistics, there are several points that could be improved.

  1. Please, check out the list of references, problems with numbering.
  2. The study lacks robust comparison to existing PDE inhibitors with similar activity. Since rather few animals have been used, it would be important to compare the A33 with older compounds head-to-head to understand its possible benefits.
  3. The manuscript is lengthy, especially the Discussion could be shortened by 30%, because the mechanisms of action was really not studied here.
  4. The abbreviations of the procedures could be avoided, making the reading simpler for all readers.

Author Response

We would like to thank this reviewer for their helpful comments and suggestions.

Please, check out the list of references, problems with numbering.

Reply:  We humbly apologize.  In response to Reviewer 1, we have added additional references and have reviewed and re-reviewed the numbering for accuracy.

The study lacks robust comparison to existing PDE inhibitors with similar activity. Since rather few animals have been used, it would be important to compare the A33 with older compounds head-to-head to understand its possible benefits.

Reply:  We completely agree with this reviewer and have these studies planned.  As our animal numbers and ability to enter the research buildings were extremely limited due to COVID-19 restrictions, we opted to move forward with the more selective inhibitor first as we were in the process of preparing a grant focusing specifically on this isozyme and preliminary data using a selective inhibitor was required. We have now included a sentence  regarding the importance of comparing A33’s efficacy with those of non-selective inhibitors and our future plans to do so. 

The manuscript is lengthy, especially the Discussion could be shortened by 30%, because the mechanisms of action was really not studied here.

Reply:  We have cut out a significant portion of Sect 3.1.  We have left the information pertaining to substrain differences in the pde4b gene and sex differences in PDE4B induction as food for thought for our readers. 

The abbreviations of the procedures could be avoided, making the reading simpler for all readers.

Reply:  We are not entirely clear to which procedure abbreviations the reviewer is referring? The only abbreviation employed is DID for Drinking-in-the-Dark.  This is a well-recognized abbreviation in the alcohol field and is often employed in both written and spoken work when referring to this specific binge-drinking procedure.